# Comparison of Bleb Morphology following PRESERFLO^®^ MicroShunt and Trabeculectomy Using Anterior Segment OCT

**DOI:** 10.3390/diagnostics13213373

**Published:** 2023-11-02

**Authors:** Somar M. Hasan, Theresa Theilig, Daniel Meller

**Affiliations:** Department of Ophthalmology, Jena University Hospital, Am Klinikum 1, 07747 Jena, Germany; theresa.theilig@med.uni-jena.de (T.T.); daniel.meller@med.uni-jena.de (D.M.)

**Keywords:** glaucoma, filtering surgery, PRESERFLO MicroShunt, MIGS, MIBS, bleb morphology, anterior segment OCT

## Abstract

Trabeculectomy has traditionally been upheld as the gold standard in glaucoma surgery, but recent advancements, including the PRESERFLO^®^ MicroShunt, have introduced less invasive techniques to mitigate complications and reduce intraocular pressure (IOP). The examination of bleb morphology post-filtering surgery is critical for modulating wound healing and ensuring long-term success. While previous evaluations of PRESERFLO-generated blebs have relied on data from trabeculectomy blebs, the differing surgical techniques used in PRESERFLO and trabeculectomy surgeries suggest potential variations in bleb morphologies. This study conducted a comparative analysis of blebs resulting from PRESERFLO and trabeculectomy procedures. Retrospective descriptive assessments using the Jenaer Bleb Grading System were performed, along with quantitative evaluation using eight-dimensional parameters utilizing anterior segment OCT. We included 93 eyes (57 following PRESERFLO, 36 following trabeculectomy). In the descriptive assessment, PRESERFLO-generated blebs exhibited fewer conjunctival cysts (3.5% vs. 22.2%, *p* = 0.007) and cavernous changes (10.5% vs. 30.5%, *p* = 0.021) compared to trabeculectomy. Quantitatively, overall bleb dimensions were comparable (maximal bleb height, width, and length, *p* > 0.05 for all). However, PRESERFLO blebs displayed a shallower (0.52 ± 0.24 vs. 0.67 ± 0.3 mm, *p* = 0.017) yet longer (4.12 ± 1.54 vs. 3.23 ± 1.64 mm, *p* = 0.024) episcleral lake. A thicker bleb wall (1.52 ± 0.46 vs. 1.10 ± 0.37 mm, *p* < 0.001) along with more posteriorly positioned blebs (distance to limbus = 6.16 ± 1.36 vs. 4.87 ± 1.34 mm, *p* < 0.001) were observed following PRESERFLO. This study illuminates the nuanced morphological differences between PRESERFLO and trabeculectomy blebs. Understanding these distinctions is vital, empowering clinicians to make informed postoperative decisions and avoid misinterpretation of bleb morphology.

## 1. Introduction

Trabeculectomy, even after several decades, remains the established gold standard in glaucoma surgery [1]. Emerging techniques like microinvasive glaucoma surgery (MIGS) and microinvasive bleb surgery (MIBS) are designed to mitigate complications post-trabeculectomy, while also effectively lowering intraocular pressure (IOP).

Assessing bleb morphology after filtering glaucoma surgery is crucial for long-term success [2,3]. Active intervention to manage scarring processes is a key aspect of postoperative care [4]. This intervention is primarily guided by intraocular pressure (IOP) levels and the bleb’s appearance. Evaluating bleb morphology is therefore highly significant in enhancing success rates. The assessment relies on established classification systems [2,3,5] initially designed for trabeculectomy blebs but also utilized in clinical practice for blebs resulting from other filtering procedures like MIBS. Traditionally, these systems have relied on biomicroscopy and photography. However, the introduction of anterior segment optical coherence tomography (AS-OCT) has provided an additional tool, revealing previously unseen structures and enabling a more objective evaluation and quantification of bleb morphology [6,7,8,9,10,11].

Variations in bleb morphology have been observed across various trabeculectomy surgical techniques [12]. It is crucial to acknowledge these variations to prevent misinterpretation of bleb morphology. Recent studies on drainage shunts indicate that blebs resulting from an ab-interno approach (such as those following XEN-Gel-Stent, AbbVie Inc., Chicago, IL, USA) appear to have smaller dimensions (width, height, and length) compared to those formed through an ab-externo approach (such as PRESERFLO) [13,14].

The PRESERFLO^®^ MicroShunt (PRESERFLO, Santen Pharmaceutical Co. Ltd., Osaka, Japan) is a relatively recent MicroShunt composed of styrene-block-isobutylene-block-styrene (SIBS) and can be implanted using an ab externo approach. Studies have demonstrated its high efficacy in reducing intraocular pressure (IOP) while maintaining a favorable safety profile [15]. However, there is limited available data concerning the characteristics of blebs formed after PRESERFLO [16,17]. To the best of our knowledge, there is no existing comparison between these blebs and those resulting from trabeculectomy. The objective of this study was to conduct a descriptive and quantitative analysis of morphological differences in blebs following PRESERFLO and trabeculectomy within a comparable group of eyes.

## 2. Materials and Methods

This is a retrospective comparative study that included eyes of patients with uncontrolled glaucoma despite maximal tolerated medications. Patients underwent one of two procedures as a stand-alone surgery between July 2019 and December 2022, and were divided into two groups: the PRESERFLO-Group, which involved the implantation of PRESERFLO with Mitomycin-C (MMC) using an ab-externo approach, and the trabeculectomy-Group, which received a fornix based trabeculectomy with MMC.

Patients with functional blebs attending follow-up visits from the 5th week post-surgery onwards were assessed using anterior segment optical coherence tomography (AS-OCT). Blebs within the initial 4 weeks post-surgery were omitted due to their rapid morphological changes during this early phase [13,18]. A functional bleb was characterized by an intraocular pressure (IOP) of ≤18 mmHg and a reduction of ≥20% from the pre-operative IOP (at the time of surgical indication), without the use of medications. Nonfunctional blebs, mainly characterized as scarred and/or flat blebs lacking specific comparable characteristics, were excluded. Eyes with prior conjunctival-affecting ocular surgeries (such as glaucoma surgery, strabismus surgery, vitrectomy, or pterygium excision) and those needing bleb revision or needling due to bleb failure were also not considered. When both eyes were eligible, the eye that underwent surgery first was chosen for the study.

Collected preoperative data included age, sex, glaucoma type, localization, IOP at the time of surgical indication, and the number of glaucoma medications (NoMs). Postoperative data included the date of surgery, date of examination, time after surgery (TaS) in days, IOP, NoM, and any postoperative complications. Each bleb was descriptively and quantitatively examined, and differences between both groups were analyzed.

### 2.1. Surgical Technique

Preoperative preparation: All glaucoma medications are discontinued for 4 weeks prior to surgery, and patients are prescribed dexamethasone eyedrops (Dexapos COMOD 1.0 mg/mL eye drops, Ursapharm, Saarbrücken, Germany) three times daily, along with oral Acetazolamide 250 mg (Glauopax^®^ 250 mg tablets, Omni-Vision, Puchheim, Germany). The dosage of Acetazolamide is adjusted based on intraocular pressure (IOP) measurements.

Implantation of PRESERFLO-MicroShunt with the application of MMC: After disinfection, a fixation suture (7-0 Vicryl, Ethicon, Somerville, NJ, USA) is inserted into the cornea at 12 o’clock, and the surgical field is exposed. A peritomy is performed superiorly over 2 clock hours with 2 radial cuts in the conjunctiva. Tenon tissue is then carefully dissected horizontally and posteriorly from the sclera, and episcleral vessels are cautiously cauterized. Two Lasik cornea shields are soaked in MMC solution (0.2 mg/mL) and placed under the tenon tissue for 3 min. After irrigating the surgical field with 20 mL balanced salt solution, the sclera is marked 3 mm posterior to the limbus, and a scleral tunnel of approximately 2 mm in length is created using the included knife. The 25-Gauge needle is inserted into the anterior chamber through the tunnel, ensuring a proper distance from the cornea and iris. The PRESERFLO-MicroShunt is then inserted into the anterior chamber through the tunnel. After confirming proper drainage, the tenon tissue is repositioned anteriorly and secured to the sclera with two interrupted sutures (10-0 Vicryl), followed by conjunctival closure with two to four interrupted sutures (10-0 Vicryl).

#### TET with Application of MMC

After disinfection, a fixation suture (7-0 Vicryl, Ethicon, Somerville, NJ, USA) is inserted into the cornea at 12 o’clock, and the surgical field is exposed. A fornix-based peritomy is performed superiorly over 2 clock hours. Tenon tissue is then carefully dissected horizontally and posteriorly from the sclera, and episcleral vessels are cautiously cauterized. Two Lasik cornea shields are soaked in MMC solution (0.2 mg/mL) and placed under the tenon tissue for 3–4 min. After irrigating the surgical field with 20 mL balanced salt solution, a 2 × 3 mm scleral lamella of about ½ to 2/3 of the sclera thickness is prepared until reaching the clear cornea, followed by a paracenteses. A small window is created in the bed of the sclera through which the anterior chamber is entered. A peripheral iridectomy is performed through the sclerotomy, and the scleral flap is closed with 4× sutures (10-0 Ethylon). The conjunctival is closed using 4 limbal sutures in a watertight manner.

Postoperative regimen: Ofloxacin eye drops (Floxal^®^ 3 mg/mL eye drops, Bausch&Lomb, Laval, QC, Canada) are administered five times daily for one month, and dexamethasone eyedrops (Dexapos COMOD 1.0 mg/mL eye drops, Ursapharm, Saarbrücken, Germany) are given every two hours during the first week, followed by five times daily starting from day 8. The dosage of dexamethasone is gradually reduced by one drop per day every four weeks. Accordingly, patients receive dexamethasone eye drops for a total of five months postoperatively. Following trabeculectomy, IOP is controlled daily and Argon-laser-assisted suture lysis is performed according to the IOP and bleb morphology.

### 2.2. Bleb Examination Using AS-OCT

A high-resolution swept-source AS-OCT (ANTERION^®^, Heidelberg Engineering GmbH, Heidelberg, Germany) was used to examine the blebs following a standardized protocol. The patient was instructed to look downwards and either temporally or nasally to expose the bleb. The upper lid was gently lifted using a cotton swab, taking care not to apply any pressure on the eye. A rectangular box, 7.5 mm wide, from the imaging module with 45 scans was positioned at the center of the stent. Active eye-tracking was turned off during the scans. Two sets of scans were performed, the first set with scans oriented parallel to the stent (mostly radial to the limbus), and the second set with scans oriented perpendicular to the stent (mostly tangential to the limbus), as shown in Figure 1. Care was taken to capture the maximum visible area of the bleb posteriorly and horizontally. This resulted in 90 scans for each bleb, which were evaluated by the same examiner (SMH) who was blinded to the clinical data.

Descriptive assessment: Images were qualitatively classified according to the Jenaer Bleb Grading System (JBGS) [19]. This grading system allows for documentation of morphological changes at three anatomical levels: the conjunctiva (C0 = no changes, C1 = intraepithelial cysts, and C2 = subconjunctival spaces), Tenon’s tissue (T0 = no changes, T1 = hyperreflective changes, T2 = hyporeflective changes, and T3 = cavernous changes), and the episcleral space (ES0 = no episcleral space and ES1 = episcleral space visible). The classification is based on comparing the changes observed in each bleb to standard images and assigning a corresponding code for each layer. For example, a bleb with subepithelial spaces (C2), hyporeflective changes in Tenon’s tissue (T2), and a visible episcleral space (ES1) would be classified as C2T2ES1.

Quantitative assessment: Images were exported and analyzed using photo editing software (Adobe Photoshop CC, Version 20.0.0, Adobe Systems Incorporated, San Jose, CA, USA). The width of the image, as provided by the AS-OCT machine, was entered into the scaling tool of the editing software. Eight parameters of each bleb were measured when applicable, as described in Table 1 and the corresponding Figure 1. These parameters were selected to assess the dimensions of the bleb in three geometric directions (height, width, and length). These included measurements of the overall bleb size (MBH, MBW, and MBL), as well as measurements of intra-bleb components, such as the dimenstions of the episcleral lake (MLH, MLW, and MLL). The thickness of the bleb wall, a well-established parameter of bleb function (BWT), and the location of the bleb (distance to the limbus, DtL) were also measured. Each of the 90 images of each bleb was evaluated for the corresponding parameters.

## 3. Results

In our study, a total of 93 eyes were analyzed, with 57 eyes in the PRESERFLO-Group and 36 eyes in the trabeculectomy-Group. Table 2 provides demographic details, as well as pre- and post-operative data. Prior to surgery, both groups showed similar characteristics without significant differences. However, after the surgery, the PRESERFLO-Group exhibited higher IOP compared to the trabeculectomy-Group (*p* = 0.036), and the percentage reduction in IOP was lower (*p* = 0.01). Apart from this, there were no notable differences in other parameters.

Table 3 presents the descriptive evaluation of blebs at various anatomical levels. The PRESERFLO-Group exhibited notably fewer conjunctival cysts (C1) in comparison to the trabeculectomy-Group (3.5% vs. 22.2%, *p* = 0.007). Similarly, cavernous changes at the tenon level (T3) were significantly less frequent in the PRESERFLO-Group (10.5% vs. 30.5%, *p* = 0.021). An example for this comparison is shown in Figure 2.

The quantitative analysis of bleb morphology results is outlined in Table 4. Overall bleb dimensions (MBH, MBW, and MBL) did not exhibit significant differences between the two groups. However, the episcleral lake following PRESERFLO was shallower than in trabeculectomy (MLH = 0.52 ± 0.24 vs. 0.67 ± 0.3 mm, *p* = 0.017) but extended further posteriorly (4.12 ± 1.54 vs. 3.23 ± 1.64 mm, *p* = 0.024). In addition, the BWT in the PRESERFLO-Group was higher than that in the trabeculectomy-Group (1.52 ± 0.46 vs. 1.10 ± 0.37, *p* = 0.00004). Also, blebs following PRESERFLO were positioned more posteriorly (DtL = 6.16 ± 1.36 vs. 4.87 ± 1.34 mm, *p* = 0.00005) in comparison to those resulting from trabeculectomy.

## 4. Discussion

Analyzing the morphology of the bleb is crucial in postoperative management after filtering surgery. Bleb evaluation, in addition to monitoring intraocular pressure (IOP), guides the active intervention to minimize scarring, a key factor in ensuring long-term success following filtering surgery. Morphological differences in blebs have been extensively documented in different trabeculectomy surgical approaches [12,20]. Recognizing these variations is essential to prevent misinterpretation, which could lead to inappropriate therapeutic choices.

The PRESERFLO-MicroShunt provides a relatively novel approach for bleb-dependent glaucoma surgery, demonstrating high efficacy and a favorable safety profile [15,21,22]. Currently, the assessment of PRESERFLO blebs heavily relies on data obtained from trabeculectomy blebs, despite the lack of concrete evidence supporting the comparability of blebs resulting from these two distinct procedures.

In our study, we observed a high prevalence of functional blebs displaying subconjunctival spaces (C2 pattern) in both groups, coupled with a lower incidence of conjunctival cysts (C1 pattern) in the PRESERFLO-Group compared to the trabeculectomy-Group. Traditionally, conjunctival cysts were considered a positive indicator of bleb functionality post-trabeculectomy [2,3]. However, the use of anterior segment optical coherence tomography (AS-OCT) enabled the detection of smaller cysts that may not always align with those visible through biomicroscopy. In this study, which exclusively included functional blebs, the elevated occurrence of conjunctival cysts in the trabeculectomy-Group might be due to the anterior drainage in this group, in contrast to the posterior drainage in the PRESERFLO-Group. This difference could be attributed to a combined subconjunctival and subtenon-based drainage in trabeculectomy, as opposed to mainly subtenon drainage in PRESERFLO. Consequently, the reduced incidence of conjunctival cysts following PRESERFLO should not be interpreted as a decrease in bleb functionality from a clinical standpoint.

At the Tenon’s layer level, all the examined blebs exhibited Tenon changes, with hyporeflective alterations (T2 pattern) being the most prevalent in both groups. Interestingly, cavernous Tenon changes (T3 pattern) were less frequently observed in the PRESERFLO-Group compared to the trabeculectomy-Group, leading to a more heterogenous morphology (vacuoles and septums) in the trabeculectomy-Group (Figure 3). These differences are likely attributable to the surgical technique and the nature of aqueous drainage following PRESERFLO. The Tenon’s layer advancement performed during bleb closure in the PRESERFLO-Group, along with a more posterior drainage (associated with the implant’s length), may have resulted in mainly subtenon drainage, leading to a more consistent tenon appearance compared to the trabeculectomy-Group. In contrast, the trabeculectomy-Group exhibited more anterior drainage and a combination of subconjunctival and subtenon drainage, which might have led to cavernous changes and increased conjunctival cysts. The presence of an episcleral lake was high and comparable in both groups, a well-known morphological feature in blebs created through an ab-externo approach [14,23].

In terms of quantitative comparison, we did not observe any differences in bleb dimensions (MBH, MBW, MBL) between the two groups. While an ab-interno approach (such as following the XEN-Gel-Stent implantation) was associated with significantly smaller blebs compared to an ab-externo technique [14], both trabeculectomy and PRESERFLO, performed through an ab-externo approach, resulted in similar bleb sizes, although eyes that underwent trabeculectomy showed lower IOP compared to PRESERFLO. The lower IOP does not seem to affect the morphology of the bleb. However, variations were noted in the dimensions of the episcleral lake between the groups. In the PRESERFLO-Group, the lake was shallower (indicated by decreased MLH) and longer (indicated by increased MLL) compared to the trabeculectomy-Group. Several factors might explain these differences. Firstly, the thicker Tenon’s layer above the episcleral lake in PRESERFLO may have resulted in higher pressure and consequently a shallower lake with more posterior extension. Secondly, the design of PRESERFLO facilitates consistent posterior outflow through the distal orifice, unlike the variability in drainage directions in trabeculectomy, which depends on the handmade scleral flap. In trabeculectomy, drainage can be directed nasally, temporally, or posteriorly. This variability in primarily anterior drainage might contribute to the increased MLH in the trabeculectomy-Group. Wide filtering surfaces (episcleral lake) were associated with successful blebs in other studies [16]. Additionally, the extended posterior dissection of Tenon’s layer from the sclera, routinely performed in PRESERFLO eyes along with posterior application of MMC, may have also contributed to the greater length of the episcleral lake.

An important finding in this study was the greater thickness of the bleb wall (BWT) in the PRESERFLO-Group compared to the trabeculectomy-Group. Similar outcomes of increased BWT were also noted in blebs following PRESERFLO compared to XEN-Gel-Stent [14]. BWT serves as a significant biomarker for bleb function, where a thicker bleb wall is generally associated with better functionality [24,25,26]. However, it is essential to exercise caution when comparing bleb thickness between PRESERFLO and trabeculectomy. One primary explanation could be the posterior drainage in PRESERFLO, located in an area with inherently higher tenon thickness. Additionally, the advancement of Tenon’s layer might have an auxiliary impact on the thickness of the bleb wall.

As anticipated based on the surgical approach, PRESERFLO-generated blebs are notably more posteriorly located compared to those resulting from trabeculectomy. This variance impacts the examination of these blebs and occasionally hampers the visibility of their posterior extension.

These differences in bleb morphology, however, were associated with lower IOP values in the trabeculectomy-Group, which comes in accordance with published data [21]. Despite this difference, bleb dimensions did not differ between both groups.

This study has several limitations. Firstly, its retrospective design and associated drawbacks pose inherent limitations. Additionally, the surgeries were conducted by two different surgeons, potentially influencing bleb morphology, although they employed identical surgical techniques. Another constraint is the subjective nature of classifying and measuring AS-OCT images, which could introduce variations. To mitigate this potential bias, a single experienced examiner meticulously performed all measurements in both groups and was blinded to clinical data. Furthermore, the changing bleb morphology over time could introduce variability into the comparison [13,16,27], as assessments were not conducted at a fixed time point. To address this concern, measurements taken within the first 4 weeks after surgery, when bleb morphology is highly variable, were excluded from the analysis. Conversely, both groups had a mean follow-up duration of over 6 months, ensuring that the comparison was made over a long time frame. Importantly, there was no significant difference in the mean TaS between the two groups. To the best of our knowledge, our study is the first to compare the morphological differences between blebs resulting from PRESERFLO and trabeculectomy. Ophthalmologists in practice should be cognizant of these distinctions to prevent misinterpretation of bleb morphology, which could potentially lead to inappropriate decisions during the postoperative phase.

## Figures and Tables

**Figure 1 diagnostics-13-03373-f001:**
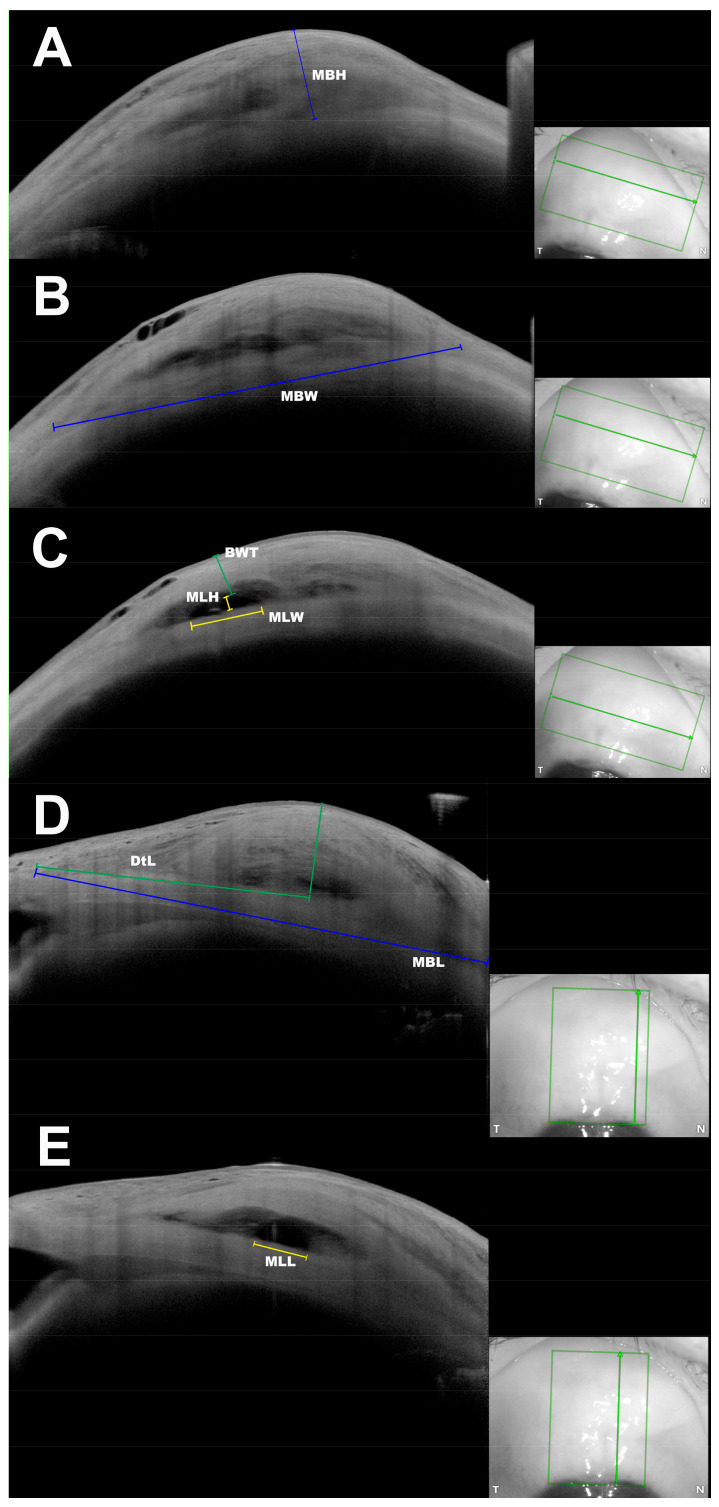
Parameters analyzed during the quantitative assessment of bleb morphology. (**A**–**C**): tangential scans, (**D**,**E**): radial scans. Abbreviations: MBH: maxim bleb height, MBW: maximum bleb width, BWT: bleb wall thickness, MLH: maximum lake height, MLW: maximum lake width, MBL: maximum bleb length, DtL: distance to limbus, MLL: maximum lake length. Green frames: captured area, green arrow: position of showed scan, T: temporal, N nasal.

**Figure 2 diagnostics-13-03373-f002:**
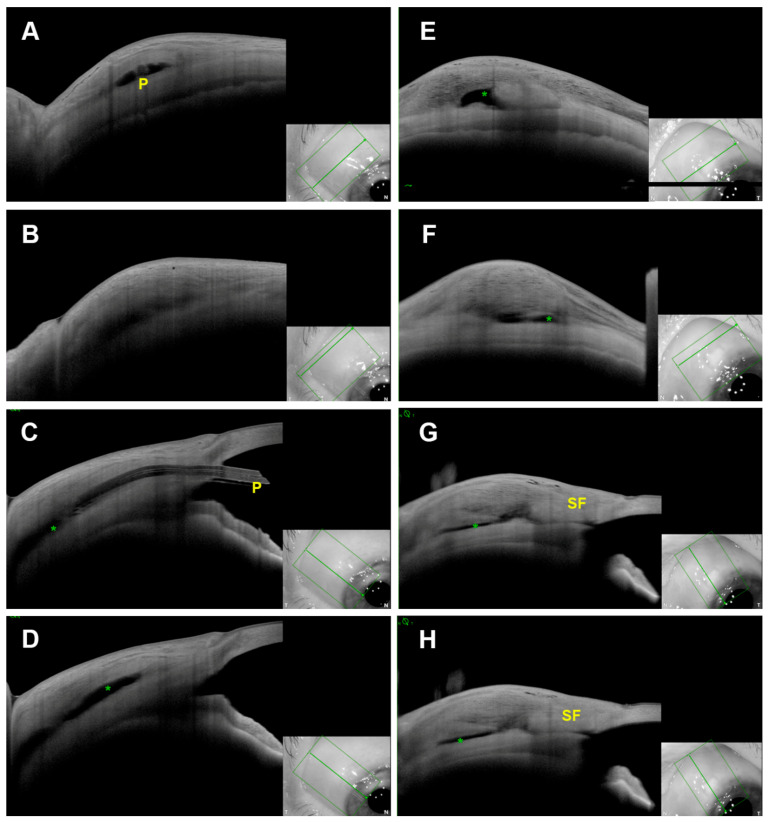
Example of two blebs: following PRESERFLO (**A**–**D**) and following trabeculectomy (**E**–**H**). P: PRESERFLO-MicroShunt, green asterisk: episcleral lake. SF: scleral flap. (**A**,**B**,**E**,**F**): tangential scans, (**C**,**D**,**G**,**H**): radial scans. Green frames: captured area, green arrow: position of showed scan, T: temporal, N nasal.

**Figure 3 diagnostics-13-03373-f003:**
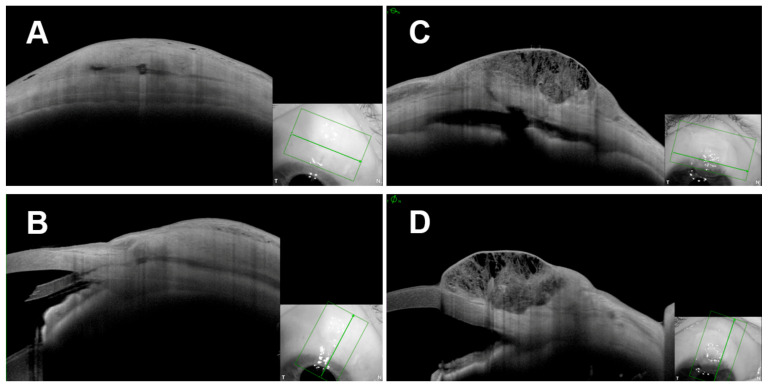
Example of different Tenons’ layer morphologies of blebs following PRESERFLO (**A**,**B**) and trabeculectomy (**C**,**D**) showing the cavernous Tenon’s layer changes (T3) following trabeculectomy compared with the more homogenous Tenon’s layer changes in the PRESESRFLO group. (**A**,**C**): tangential scans, (**B**,**D**): radial scans. Green frames: captured area, green arrow: position of showed scan, T: temporal, N nasal.

**Table 1 diagnostics-13-03373-t001:** Definition of quantitative parameters measured using AS-OCT. Abbreviations: ES1: pattern episcleral 1 of the Jenaer Bleb Grading System (JBGS).

Parameter	Abbreviation	Description	Remarks
Maximum Bleb Height Figure 1A	MBH	The maximum height of the bleb seen in the tangential scans, measured as the maximum perpendicular distance from the sclera to the first reflex at the conjunctiva.	
Maximum Bleb Width Figure 1B	MBW	The maximum width of the bleb seen in tangential scans, measured as a direct line between two points: begin of changes in tenon thickness nasally to end of tenon changes temporally.	If the whole width of the bleb could not be captured in a single image, the maximum visible width was measured.
Maximal Bleb Length Figure 1D	MBL	The maximum posterior extension of the bleb seen in radial scans, measured as a direct line between two points: from the first changes in tenon thickness anteriorly to the last visible tenon change posteriorly. If the bleb extended over the cornea, measurement was started at the level of the scleral spur.	If the whole length of the bleb could not be captured in a single image, the maximum visible length was measured.
Maximum Lake Height Figure 1C	MLH	The maximum height of the episcleral lake (ES1 according to JBGS) in the tangential scans, measured as maximum perpendicular distance from the inferior to the superior edge of the episcleral lake.	MLH was measurable only in blebs showing the pattern ES1.
Maximum Lake Width Figure 1C	MLW	The maximum width of episcleral lake seen in tangential scans, measured as a direct line between two points: begin of episcleral lake nasally to its end temporally.	MLW was measurable only in blebs showing the ES1-Pattern.
Maximal Lake Length Figure 1E	MLL	The maximum posterior extension of the episcleral lake seen in a radial scan, measured as a linear distance between two points: begin of the episcleral lake anteriorly to its end posteriorly.	If the whole length of the episcleral lake could not be captured in a single image, the maximum visible length was measured.
Bleb Wall Thickness Figure 1C	BWT	Minimal thickness of the bleb wall at the scan with the MLH, measured as the minimal perpendicular distance between the end of the episcleral lake and the first reflex at the conjunctiva.	BWT was measurable only in blebs showing the ES1-Pattern.
Distance to Limbus Figure 1D	DtL	The linear distance between two points: point of corneal surface corresponding to the scleral spur and point of scleral surface corresponding to the highest point of the bleb in radial scans.	

**Table 2 diagnostics-13-03373-t002:** Demographic and post-operative data. Abbreviations: NoM: number of medications, TaS: time after surgery, IOP: intraocular pressure. Significant *p*-Values were marked in bold.

Parameter	PRESERFLO-Group	Trabeculectomy-Group	*p* Value
Age (years)	68.1 ± 12.9	70.3 ± 9.0	0.38
Sex: Male (%)	25 (43.8%)	14 (38.9%)	0.64
Preoperative IOP (mmHg)	25.8 ± 10.6	26.2 ± 8.7	0.82
Preoperative NoM	2.9 ± 1.3	2.8 ± 1.2	0.67
Type of Glaucoma: n (%) - Primary open anlge glaucoma - Pseudoexfoliation glaucoma - Uveitic glaucoma - Others	- 39 (68.4%) - 7 (12.3%) - 6 (10.5%) - 5 (8.9%)	- 25 (69.4%) - 8 (22.2%) - 1 (2.8%) - 2 (5.6%)	0.22
TaS (days)	213.6 ± 171	270.6 ± 215	0.16
Postoperative IOP (mmHg)	11.5 ± 3.3	10.0 ± 3.4	**0.036**
IOP reduction (mmHg)	14.2 ± 10.3	16.2 ± 7.9	0.33
IOP reduction (%)	50.2 ± 17.7	59.7 ± 15.8	**0.01**

**Table 3 diagnostics-13-03373-t003:** Descriptive comparison of bleb morphology according to the Jenaer Bleb Grading System (JBGS). C0: no conjunctival changes, C1: conjunctival cysts, C2: subconjunctival spaces, T0: no tenon changes, T1 hyperreflective tenon changes, T2: hyporeflective tenon changes, T3: cavernous tenon changes, ES1: episcleral lake visible, chi-square test for all cases. Significant *p*-Values were marked in bold.

Morphological Pattern	PRESERFLO-Group	Trabeculectomy-Group	*p* Value
C0	5 (8.8)	2 (5.5)	0.5
C1	2 (3.5)	8 (22.2)	**0.007**
C2	46 (80.7)	26 (72.2)	0.086
T0	0 (0)	0 (0)	-
T1	16 (28.0%)	5 (13.8)	0.084
T2	32 (56.1)	20 (55.5)	0.727
T3	6 (10.5)	11 (30.5)	**0.021**
ES1	50 (87.7)	28 (77.8)	0.1

**Table 4 diagnostics-13-03373-t004:** Quantitative assessment and comparison of bleb morphology. Significant *p*-Values were marked in bold.

Parameter (mm)	PRESERFLO-Group	Trabeculectomy-Group	*p* Value
MBH	2.17 ± 0.47	2.15 ± 0.43	0.83
MBW	10.67 ± 2.23	11.23 ± 2.25	0.24
MBL	9.59 ± 1.57	9.9 ± 1.56	0.39
MLH	0.52 ± 0.24	0.67 ± 0.3	**0.017**
MLW	3.69 ± 1.95	3.51 ± 2.22	0.72
MLL	4.12 ± 1.54	3.23 ± 1.64	**0.024**
BWT	1.52 ± 0.46	1.10 ± 0.37	**0.00004**
DtL	6.16 ± 1.36	4.87 ± 1.34	**0.00005**

## Data Availability

Data are available on request from the authors.

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
