# Peer review of "Comparison of Bleb Morphology following PRESERFLO® MicroShunt and Trabeculectomy Using Anterior Segment OCT"

_diagnostics, 2023, doi:10.3390/diagnostics13213373_

Round 1
Reviewer 1 Report
Comments and Suggestions for Authors
I had the pleasure to review this manuscript.
The shape of the filtration bleb may change over time. It is better to decide when to measure. We believe that the high variability in TaS (Time after surgery) is a limitation of this study.
Reviewer 2 Report
Comments and Suggestions for Authors
This is an interesting study reporting clinical results after implanting a MIGS device for lowering IOP in glaucoma patients.
I have few minor comments pertaining to this manuscript:
1) page 3 line 116: Please correct the following: "…followed by a paracenteses."
2) page 9 line 208: Please correct the following: "Analyzing the morphology of the bleb is crucial in the postoperative management…"
3) page 10 line 234: Please correct the following: "At the Tenon's layer level,…"
4) page 10 line 240: Please correct the following: "The Tenon's layer advancement…"
5) page 10 lines 249-251: Please correct the following: "Figure 3. example of different Tenon's layer morphologies…" and "…showing the cavernous Tenon's layer changes…" and "…homogenous Tenon's layer changes…"
6) page 11 line 263: Please correct the following: "…the thicker Tenon's layer above the episcleral lake…"
7) page 11 line 270: Please correct the following: "…of Tenon's layer from the sclera…"
8) page 11 line 280: Please correct the following: "…the advancement of Tenon's layer…"
Reviewer 3 Report
Comments and Suggestions for Authors
Please add an important paper:
Barbera MI et al, Plos One.2023 Jun 8;18(6):e0286884.doi: 10.1371/journal.pone.0286884. eCollection 2023. The paper ist carefully prepared and interesting, but I think that you need to add some other relevant papers, which have been published in 2023.
